# Balance between Pro- and Antifibrotic Proteins in Mesenchymal Stromal Cell Secretome Fractions Revealed by Proteome and Cell Subpopulation Analysis

**DOI:** 10.3390/ijms25010290

**Published:** 2023-12-25

**Authors:** Maria Kulebyakina, Nataliya Basalova, Daria Butuzova, Mikhail Arbatsky, Vadim Chechekhin, Natalia Kalinina, Pyotr Tyurin-Kuzmin, Konstantin Kulebyakin, Oleg Klychnikov, Anastasia Efimenko

**Affiliations:** 1Faculty of Medicine, Lomonosov Moscow State University, 27/1, Lomonosovskiy Av., 119192 Moscow, Russia; coolebyakina@gmail.com (M.K.); natalia_ba@mail.ru (N.B.); butuzova_dasha@mail.ru (D.B.); algenubi81@mail.ru (M.A.); v-chech@mail.ru (V.C.); n_i_kalinina@mail.ru (N.K.); tyurinkuzmin.p@gmail.com (P.T.-K.); konstantin-kuleb@mail.ru (K.K.); 2Institute for Regenerative Medicine, Medical Research and Educational Center, Lomonosov Moscow State University, 27/10, Lomonosovskiy Av., 119192 Moscow, Russia; 3Faculty of Biology, Lomonosov Moscow State University, 1-12, Leninskie Gory, Lomonosovskiy Av., 119991 Moscow, Russia; oklych@yahoo.co.uk

**Keywords:** multipotent mesenchymal stromal cells, secretome, extracellular vesicles, proteomics, fibrosis, fibroblasts, cell differentiation, myofibroblasts, single cell RNA sequencing, cell subpopulation

## Abstract

Multipotent mesenchymal stromal cells (MSCs) regulate tissue repair through paracrine activity, with secreted proteins being significant contributors. Human tissue repair commonly results in fibrosis, where fibroblast differentiation into myofibroblasts is a major cellular mechanism. MSCs’ paracrine activity can inhibit fibrosis development. We previously demonstrated that the separation of MSC secretome, represented by conditioned medium (CM), into subfractions enriched with extracellular vesicles (EV) or soluble factors (SF) boosts EV and SF antifibrotic effect. This effect is realized through the inhibition of fibroblast-to-myofibroblast differentiation in vitro. To unravel the mechanisms of MSC paracrine effects on fibroblast differentiation, we performed a comparative proteomic analysis of MSC secretome fractions. We found that CM was enriched in NF-κB activators and confirmed via qPCR that CM, but not EV or SF, upregulated NF-κB target genes (*COX2*, *IL6*, etc.) in human dermal fibroblasts. Furthermore, we revealed that EV and SF were enriched in TGF-β, Notch, IGF, and Wnt pathway regulators. According to scRNAseq, 11 out of 13 corresponding genes were upregulated in a minor MSC subpopulation disappearing in profibrotic conditions. Thus, protein enrichment of MSC secretome fractions and cellular subpopulation patterns shift the balance in fibroblast-to-myofibroblast differentiation, which should be considered in studies of MSC paracrine effects and the therapeutic use of MSC secretome.

## 1. Introduction

Tissue repair in humans is ensured using the coordinated functioning of cells of different types, with myofibroblasts playing a pivotal role. Myofibroblasts mostly originate from activated stromal fibroblasts and provide healing in the injury site through deposition and contraction of the extracellular matrix [1,2]. Excessive formation of myofibroblasts and dysregulations in their elimination results in fibrosis—a pathological state characterized by the excessive accumulation of the extracellular matrix and the formation of fibrotic tissue replacing normal tissues [3]. An in-depth study of mechanisms regulating fibroblast-to-myofibroblast differentiation is an essential step both to prevent the progression of fibrosis and to find new targets for the treatment of diseases associated with fibrosis and, therefore, represents a global scientific goal of tissue regeneration.

Multipotent mesenchymal stromal cells (MSCs) are the key coordinators of all cellular processes occurring in the stroma. The regulatory role of MSCs is crucial during tissue repair and is primarily provided by the secretion of various paracrine factors, which attract immune cells to the injury site, modulate inflammatory reactions, stimulate nerve and blood vessel outgrowth, and regulate myofibroblast formation and functioning [4,5]. Distinct fractions of MSC-secreted proteins are shown to effectively prevent and reverse fibroblast differentiation in in vitro and in vivo studies, which makes MSC secretome a promising source of therapeutic tools for the prevention and treatment of fibrosis-associated diseases. However, the MSC secretome represents a complex molecular cocktail containing both pro- and antifibrotic factors, which contributes to the inconsistency of treatment outcomes [6,7].

In this work, we aimed to identify and semi-quantitatively estimate the abundance of proteins within the human adipose tissue MSC secretome, which is potentially involved in the regulation of the differentiation of fibroblasts into myofibroblasts. Our special focus was on the comparative proteome analysis of MSC secretome subfractions enriched by extracellular vesicles (EV) of soluble protein factors (SF), as these subfractions demonstrated different effects on fibrotic processes [7]. Based on the results, we proposed several mechanisms of MSC secretome potential involvement in the regulation of fibrosis by affecting NF-κB, transforming growth factor-beta (TGF-β), Wnt, Notch, and insulin growth factor (IGF) signaling pathways, as well as identified proteins mediating these mechanisms. In addition, using single-cell RNA sequencing (scRNAseq), we found that primary human adipose tissue-derived MSCs contain a subpopulation overrepresenting the transcripts of many factors revealed by proteomic analysis as potentially involved in the antifibrotic effect of MSC secretome fractions. Furthermore, profibrotic conditions were able to change these expression patterns, resulting in a small MSC subpopulation disappearance.

These effects must be considered in the further study of MSC involvement in the regulation of fibrosis and the development of novel approaches to treat fibrotic diseases.

## 2. Results

### 2.1. Distinct Effect of MSC Secretome Fractions on Fibroblast-to-Myofibroblast Differentiation Induced by TGF-β1

In our previous work, we demonstrated that MSC secretome subfractions (EV and SF) can significantly inhibit TGF-β1-induced differentiation of primary human dermal fibroblasts into myofibroblasts [6]. However, we revealed that the antifibrotic effect of the CM fraction from the same samples of MSC secretome was much weaker (Figure 1). Thus, we demonstrated that EV and SF subfractions of the MSC secretome sufficiently prevented TGF-β1-induced formation of stress fibrils rich in alpha-smooth muscle protein (αSMA) in human dermal fibroblasts, while the CM fraction of the MSC secretome did not have such a pronounced effect.

To reveal the reasons for differences in the effects of MSC secretome fractions on fibroblast-to-myofibroblast differentiation, we separated the CM fraction into EV and SF subfractions (Figure 1a) and controlled the enrichment with exosome protein markers in these subfractions using Western blotting (Appendix A). EV subfraction was pronouncedly enriched in proteins located inside the vesicles (such as beta-tubulin and HSP70), while the proteins located on the periphery of the vesicles (CD81 and CD63) were found in comparable amounts in the SF subfraction. This marker separation is consistent with the theoretical principles of the ultrafiltration procedure and confirms the integrity of the vesicles. It is important to note that the molecular masses of the detected proteins corresponded to the predicted values, indicating that no sample degradation occurred during the isolation procedure.

We suggested that the isolation of EV and SF protein subfractions could lead to enrichment in components, preventing fibroblast-to-myofibroblast differentiation, along with depletion in some components promoting it. To check this hypothesis, we performed a semi-quantitative proteomic analysis of MSC secretome fractions. Since MSCs cultured in standard conditions secrete a relatively small amount of proteins, and primary MSCs have a limited number of cell divisions, to facilitate in-depth proteomics, we employed a strategy of using immortalized hTERT MSCs (ASC52telo and ATCC). The effects of these cells mediated by secretome components were comparable with primary human adipose tissue-derived MSCs in an in vitro model of TGF-β1-induced differentiation of fibroblasts into myofibroblasts [6].

### 2.2. Fractionation Alters Quantitative Protein Content of MSC Secretome

We performed mass spectrometry-based proteomics analysis of isolated fractions of the human MSC secretome. In all experiments, we were able to identify 548 proteins in MSC secretome samples in total. Hierarchical clustering analysis for each of the three separate biological repeats demonstrated grouping samples of the same type that independently confirms the sample-to-sample reproducibility (see heat map, Appendix A). It is important to highlight that the EV subfraction is enriched in membrane proteins and is profoundly different from both the CM fraction and SF subfraction. This result is a solid confirmation of the high efficiency in the separation of vesicles from soluble protein factors through ultrafiltration.

According to Gene Onthology annotation [8], 79% (432 out of 548) of identified proteins are secreted (Appendix A), and their localization is designated as extracellular space (GO:0005615) and/or in the extracellular matrix (GO:0031012). This proves that the biological effects of the studied fractions are mainly mediated through the exoproteome.

Further comparative proteomic analysis of MSC secretome fractions revealed that 85.7% (367 out of 432) were identified in each sample (Figure 2a), with only a few secreted proteins unique to EV and SF subfractions. The separation of EV from SF increased the depth of the analysis, which allowed the identification of an additional 41 proteins. The latter was not detected in the CM fraction (13 proteins were detected in the EV subfraction only, 20 were found in SF only, and 8 proteins were found both in EV and SF subfractions). Surprisingly, we identified 10 proteins that were unique to the CM fraction. These proteins were presumably depleted through EV and SF subfraction preparation.

Quantitative analysis of these 367 proteins found in each sample showed that they could be split into three main groups according to their abundance (Figure 2b). Lists of secreted proteins whose content differs among MSC secretome fractions are presented in Appendix A. To verify the results of our semi-quantitative proteomic analysis, we performed a Western blotting assay for the set of highly represented proteins and confirmed the observed differences (Appendix A).

Thus, we found that EV and SF subfractions of the MSC secretome were enriched in 134 proteins and depleted in 189 proteins compared to the CM fraction (Figure 2). Taking together the results of qualitative and semiquantitative analysis, we can conclude that the separation of the secretome into EV and SF subfractions results in an enrichment of 175 proteins and a depletion of 199 proteins compared to the initial fraction (CM) proteome (Appendix A).

### 2.3. Proteome Data Analysis of MSC Secretome Fractions Revealed Proteins Involved in NF-κB, TGF-β1, Wnt, Notch, and IGF Signaling Pathways

#### 2.3.1. Total MSC Secretome Fraction, Not Able to Efficiently Prevent Myofibroblast Differentiation, Is Enriched in Proteins Involved in NF-κB Pathway Activation

To reveal factors potentially impeding the ability of MSC secretome to promote myofibroblast differentiation, we analyzed 199 proteins underrepresented in EV and SF subfractions compared to the CM fraction (Figure 2, pink). Gene enrichment analysis using gProfiler (version e107_eg54_p17_bf42210; [9]) showed genes annotated as participating in intracellular signaling cascades of immune system cells, as well as regulating transport and uptake of insulin-like growth factors (IGFs) by affecting IGF-binding proteins (IGFBPs) [10] (Appendix A). Literature analysis of proteins abundant in the CM fraction showed an additional six proteins involved in inflammatory response by activating NF-κB in target cells. Data on the presence of the abovementioned protein groups in MSC secretome fractions are shown in Table 1.

Since our results showed the possible involvement of NF-κB pathway activation in the effects of the CM fraction, we investigated NF-κB target genes in our model of TGF-β1-induced fibroblast-to-myofibroblast differentiation using qPCR. We demonstrated that the CM fraction, but not EV or SF subfractions, when added simultaneously with TGF-β1, enhanced the expression of NF-κB target genes in human dermal fibroblasts (Figure 3).

#### 2.3.2. Subfractions Preventing Myofibroblast Differentiation Are Enriched in Proteins Involved in the Regulation of TGF-β1, Wnt, Notch, and IGF Signaling Pathways

Similarly, we performed functional annotation of 175 proteins abundant in EV and SF subfractions, which efficiently prevent myofibroblast differentiation compared to the CM fraction (Figure 2, green). We found nine genes annotated as involved in the regulation of transport and uptake of insulin-like growth factors (IGFs) by acting on IGF-binding proteins (REAC:R-HSA-38142, Regulation of IGF transport and uptake by IGFBPs) and factors involved in the regulation of TGF-β1, Wnt, Notch, and IGF signaling pathways (Table 2).

Thus, EV and SF subfractions are enriched with 17 proteins that may regulate fibroblast differentiation by interfering with either IGF, Notch, TGF-β, or Wnt signaling pathways.

### 2.4. Gene Set Revealed through Proteomic Analysis Is Affected in a Subpopulation of MSCs under Profibrotic Conditions

To investigate which potential antifibrotic genes, revealed in our proteomic experiments, are affected in MSC in response to fibrotic stimuli, we employed our previously obtained scRNAseq datasets of adipose tissue-derived MSCs in a model of the profibrotic environment [28]. After initial data processing, all cells were divided into seven clusters, each representing a distinct subpopulation according to their transcriptional profile (Figure 4). Thus, cluster 0 includes cells with upregulated genes associated with the actin cytoskeleton regulation and formation of tight junctions (e.g., *TPM2*, *ACTA2*, *CALD1*, and *TAGLN*). Cluster 1 corresponds to MSCs in a basal undifferentiated state; cells of this cluster are characterized by increased expression of genes associated with oxidative phosphorylation and synthetic processes (*SNHG29*, *RPS12*, *ATP5F1E*, *HSPB7*, *HSPB6*, *ATP5MD*, *RPL12*, and *RPS27L*). Cluster 2 includes cells in the G2M and S phases of the cell cycle; cells of this cluster express genes associated with the cell cycle (*TOP2A*, *TYMS*, *MKI67*, *CENPF*, *TUBB4B*, *TUBA1B*, and *PTTG1*). Cluster 3 represents cells with increased expression of genes associated with extracellular matrix degradation (*MMP2*, *CTSK*, *LUM*, and *CLU*). Cluster 4 includes cells expressing extracellular matrix genes (*POSTN*, *COL5A1*, and *FBN1*). Cluster 5 cells are characterized by increased expression of genes associated with contractility (*MYH11*, *ACTA2*, *ACTG2*, *MYLK*, and *CALD1*) and, therefore, potentially displaying a smooth muscle cell-like phenotype. Cluster 6 cells are characterized by the expression of genes associated with the regulation of the immune response (*CXCL8*, *LYZ*, *CCL2*, *CCL3*, *CD74*, and *FCER1G*).

Surprisingly, virtually each potential antifibrotic gene (11 out of 13 genes determined through scRNAseq and coding for the factors revealed using proteomic analysis) identified in transcriptomic profiles was upregulated in the same minor subpopulations denoted as cluster 4 (Table 2), thus representing its characteristic gene profile. Details on the expression of these genes in each cluster are presented in Appendix A.

Moreover, under profibrotic conditions, cluster 4 is significantly reduced (Figure 4): in the control sample, the proportion of cells of the total subpopulation per cluster 4 was 3.85%, and after four days of cultivation under profibrotic conditions, it significantly decreased to 0.86%. The number and proportion of cells in each cluster for both samples are presented in Appendix A. Therefore, the genes of factors revealed using proteomic findings match the characteristic pattern of genes of MSC subpopulations sensitive to profibrotic stimuli.

## 3. Discussion

The secretome produced by regulatory cells is a complex mixture of components, often having oppositely directed actions. The resulting effect of this mixture is determined using the balance of molecules with different effects, as well as different secretome-producing cell subpopulations. In relation to tissue healing, it is well known that the MSC secretome contains both pro- and antifibrotic molecules [29]. Moreover, since MSCs are susceptible to plenty of environmental signals, para- and autocrine effects can alter the structure of the total tissue MSC population and its integral properties.

In our work, using an in vitro model, we tested subfractions of adipose tissue-derived MSC secretome proteins for their ability to prevent morphological changes accompanying the differentiation of fibroblasts into myofibroblasts. Compared to the total fraction of the MSC secretome, which lacks such a pronounced ability, its subfractions, enriched in extracellular vesicles (EV) and soluble protein factors (SF), demonstrated an evident antifibrotic effect in a given model. This phenomenon can be explained using the comparative enrichment of the EV and SF subfractions in sets of proteins with antifibrotic properties compared to the CM fraction and, in addition to this, through the enrichment of the CM fraction with profibrotic proteins. Using in-depth analysis of proteome, we identified a total of 548 proteins in MSC secretome samples, and the proven separation primarily alters the quantitative content of fractions studied.

The CM fraction, which did not demonstrate a pronounced antifibrotic effect, was enriched in proteins involved in the activation of the proinflammatory transcription factor NF-κB. Among these proteins, there is SDF-1, a ligand for the CXCR4 receptor, which activates NF-κB [16]; coagulation factor X, a serine endopeptidase, which is able to activate NF-κB through interaction with protease-activated receptor-1 (PAR-1) [12]; IL-6 whose ability to activate NF-κB has been well studied [13]; AKR1C3, which is also able to activate NF-κB [11]; spondin-2, and MFAP4. As shown by Yang et al., genetic knockout of spondin-2 leads to reduced NF-κB activation in a kidney fibrosis model [15]; similar results have also recently been obtained for MFAP4 [14]. We confirmed that the CM fraction, but not EV or SF subfractions, leads to the upregulation of NF-κB transcriptional targets in human dermal fibroblasts. NF-κB activation plays a significant role in fibrosis-related disease pathogenesis, and inhibition of this signaling mediator in vivo prevents fibrosis development and in vitro impedes myofibroblast differentiation [30]. Isolation of EV and SF subfractions shifts the balance of NF-κB regulators, which, therefore, alters NF-κB-driven upregulation of proinflammatory and profibrogenic cytokines in target cells. Our results confirm that the procedure for MSC secretome fraction separation changes the content of molecules, triggering the inflammatory signaling cascade, which should be considered in biomedical applications.

EV and SF fractions, both having an antifibrotic effect in our model, turned out to be enriched with regulators of Wnt (DKK3, ISLR, and PTK7), IGF (CKAP4, FAM20C, GAS6, LTBP1, MMP1, NOV, PAPPA, PRSS23, STC2, and TNC), and Notch (SNED1 and NOV) pathways. Since each of these pathways is a regulator of stromal cell differentiation, the production of their regulators can reflect the overall extent of cell differentiation potency. It should be emphasized that, although for each identified potential antifibrotic protein, there is discrete evidence of its interaction with Wnt, IGF, and/or Notch signaling pathways, our results allow for considering them as an entire protein pattern and can further be used to estimate their integral effect. The existence of the molecular pattern we discovered was confirmed for a minor subpopulation of MSCs, which highlights its physiological significance and vastly enhances the importance of our findings. The fact that our proteomic results on immortalized MSC cell secretomes are highly convergent with the scRNAseq data obtained on the primary MSC population indicates that the in vitro models we used are of great relevance.

TGF-β is a crucial profibrotic agent secreted by a wide range of cells and acts virtually on all stromal cells [31]. Here, we have ensured that MSC secretome subfractions EV and SF, which prevent TGF-β-induced differentiation, are enriched in proteins capable of inhibiting the response of target cells to TGF-β—particularly, with latent TGF-β-binding proteins 1 and 2, caveolin-1, which promotes the internalization and degradation of the first type TGF-β receptor (TGF-βRI) [17], and MXRA-5, which prevents excessive upregulation of collagen and fibronectin expression in response to exogenous TGF-β1 [22].

There is a known shift in MSC secretome composition in response to TGF-β towards a greater representation of profibrotic proteins [32]. Also, like many other cells of mesenchymal origin (fibroblasts, pericytes, smooth muscle cells, etc.), MSCs themselves are able to differentiate into myofibroblasts after stimulation by TGF-β [33,34]. We have previously described the heterogeneity of MSCs in the context of their response to TGF-β [28]. The subpopulation that diminished in response to TGF-β in our model of profibrotic conditions demonstrated the highest level of gene transcripts for potentially antifibrotic factors identified in our proteomic experiments.

Bringing to attention the fact that MSCs can secrete TGF-β on their own, our findings display that MSCs, partially in an autocrine manner, are able to regulate themselves along with other cells. This, under normal circumstances, ensures the sustainability of tissue stroma in the presence of minor stimuli, but in response to crude injury-induced factors, it can generate a positive feedback loop contributing to the pathogenesis of fibrotic diseases.

## 4. Materials and Methods

### 4.1. Cell Culture

Primary cell lines of MSCs from adipose tissue of healthy donors were obtained from the collection of the biobank of the Institute for Regenerative Medicine, Medical Research, and Educational Center, Lomonosov MSU (within the frame of the Lomonosov MSU Project “Noah’s Ark”), collection ID: MSU_MSC_AD (https://human.depo.msu.ru accessed on 15 February 2023). Primary cell lines of human dermal fibroblasts were obtained from the same biobank, collection ID: MSU_FB. Immortalized human MSCs from adipose tissue were purchased from ATCC (hTERT ASC52telo).

The collections of biomaterials from donors were created and replenished in accordance with the permission of the institutional local ethical committee (Ethics Committee of the Lomonosov Moscow State University Research Center, IRB00010587) (# 4, 4 June 2018), with the receipt of voluntary informed consent from all donors.

Cells were cultured at 37 °C and 7% CO_2_. The composition of the culture medium for ASC52telo cells was the following: DMEM with low glucose content (DMEM LG, Gibco, Waltham, MA, USA) with the addition of 10% fetal bovine serum (Gibco, Thermo Fisher Scientific, Waltham, MA, USA), 2 mM L-alanyl-L-glutamine (Thermo Fischer Scientific, Waltham, MA, USA), 1 mM pyruvate (Thermo Fischer Scientific, Waltham, MA, USA), and 100 U/mL penicillin/streptomycin (Gibco, Thermo Fisher Scientific, Waltham, MA, USA). Human adipose tissue MSCs were cultured in Advance Stem Cell Basal Medium (HyClone, Cytiva, Washington, DC, USA) containing 10% Advance Stem Cell Growth Supplement (HyClone, Cytiva, Washington, DC, USA) and 100 U/mL penicillin/streptomycin (Gibco, Thermo Fisher Scientific, Waltham, MA, USA). Human dermal fibroblasts were cultured in DMEM LG supplemented with 10% fetal bovine serum (FBS) and 1% penicillin-streptomycin (Gibco, Thermo Fisher Scientific, Waltham, MA, USA). Cells were passaged when reaching ≈80% confluency. All experiments were performed on primary MSCs no later than 6 passages and on fibroblasts no later than 12 passages.

### 4.2. Conditioning and Isolation of Secretome Fractions

To obtain MSC secretome fractions, cells were grown until they reached 80–90% monolayer confluency. To reduce the effects of media content, cells were washed twice with Hanks’ buffer solution (Paneco, Moscow, Russia). Then, MSCs were deprived in a deprivation and conditioning medium (DMEM LG without the addition of phenol red (Gibco, Thermo Fischer Scientific, Waltham, MA, USA); 2 mM L-alanyl-L-glutamine (Thermo Fischer Scientific, USA); 1 mM pyruvate (Thermo Fischer Scientific, Waltham, MA, USA); and 100 U/mL penicillin/streptomycin (Gibco, Thermo Fischer Scientific, Waltham, MA, USA)) for 24 h. After the deprivation, the medium was collected and centrifuged for 10 min at 2000× *g* at 4 °C to remove cell debris. Then, we used ultracentrifugation in centrifuge filters (Sartorius, Göttingen, Germany) with a pore diameter of 10 kDa to obtain a fraction of conditioned medium (CM); to separate extracellular vesicles (EV), filters with 300 kDa cut-off pores were used, after which the filtrate was concentrated using a 10 kDa filter to obtain a fraction of soluble factors (SF). All fractions were concentrated 200–300 times through ultracentrifugation. Samples of isolated secretome fractions were stored at −80 °C.

### 4.3. In Vitro Model of Fibroblast-to-Myofibroblast Differentiation

To study the effect of MSC secretome fractions on the induction of myofibroblast differentiation, an in vitro model of TGF-β1-induced differentiation of fibroblasts into myofibroblasts was used. For this, primary human dermal fibroblasts (8–12 passages) were seeded in culture plates in a complete growth medium at a rate of 15,000/cm^2^. After 24 h, the plates were washed with DMEM LG (Gibco) and left in the second change of DMEM LG for overnight deprivation. After deprivation, appropriate solutions were added to the cells in each group to induce differentiation: negative control—DMEM LG; positive control—DMEM LG + 5 ng/mL TGF-β1 (Cell Signalling, Danvers, MA, USA); EV—DMEM LG + 5 ng/mL TGF-β1 + EV subfraction of MSC secretome; SF—DMEM LG + 5 ng/mL TGF-β1 + SF subfraction of MSC secretome; CM—DMEM LG + 5 ng/mL TGF-β + CM fraction of MSC secretome. After 4 days of incubation, the cells were used for further analysis.

### 4.4. Immunocytochemistry

Cells were fixed with 4% paraformaldehyde in phosphate-buffered saline (Paneco, Moscow, Russia) for 10 min at room temperature and permeabilized with 0.1% Triton X-100 for 10 min. Non-specific binding was blocked with 10% normal goat serum (Abcam, Cambridge, UK) in 1% bovine serum albumin (Paneco, Moscow, Russia), and cells were stained with antibodies to alpha-actin (ab32575; Abcam; dilution 1/100), Alexa 594-phalloidin (A12381; Molecular probes, Eugene, OR, USA), or non-specific rabbit IgG (NSC-2025; Santa Cruz Biotechnology, Dallas, TX, USA) overnight at 4 °C. Staining with secondary antibodies conjugated to Alexa 488 or 594 (#A11034, #A21203, Invitrogen Waltham, MA, USA) was performed at room temperature in the dark for 1 h. The nuclei were stained with DAPI (D9542, Sigma-Aldrich, St. Louis, MO, USA). Images from at least 4 representative fields of view per well were obtained using an inverted microscope with a fluorescent module, Leica DMi8, and Leica DFC7000 T cameras (Leica Microsystems GmbH, Wetzlar, Germany), followed by processing with LasX (Leica Microsystems GmbH, Wetzlar, Germany) and FIJI (GitHub Inc., San Francisco, CA, USA).

### 4.5. Sample Preparation for Proteomic Analysis

Proteomic analysis was performed in triplicate on secretome fraction samples from independently cultured cells. For each experiment, 3–5 technical replicas were used for each secretome fraction. To prepare samples for mass-spectrometric analysis, we used a bulk trypsinolysis method in a solution similar to that described in [35] using surfactant RapiGest (Waters, Milford, MA, USA) as a solubilizing agent and modified porcine trypsin (Sequencing Grade Modified Trypsin, Promega, Madison, WI, USA). Protein concentration in samples before trypsinolysis was leveled off through densitometry of SDS PAGE-separated samples stained with Coomassie Blue silver G-250 [36]. Cysteines in protein samples were reduced with 2 mM of tris (2-carboxyethyl) phosphine (TCEP, Pierce) and alkylated with methyl methanethiosulfonate (MMTS, Pierce Waltham, MA, USA). Tryptic peptides were purified using ZipTip columns (C-18; Merck Millipore, Burlington, MA, USA); the eluate was dried and redissolved in a 0.1% formic acid solution.

LC–MS/MS analysis of peptides was carried out at the Skolkovo Shared Use Center. For peptide separation, Ultimate 3000nano UPLC was used, and ESI was coupled to an MS detector (timsTOF PRO Brucker, Billerica, MA, USA). The protocol for sample separation was as follows: 1 µg of peptide digest was loaded onto an Acclaim PepMap C18 100 Å precolumn (0.5 mm × 3 mm, with a particle size of 5 µm, Thermo Fischer Scientific, Waltham, MA, USA) at a flow rate of 10 µL/min with isocratic mobile phase A (2% acetonitrile and 0.1% formic acid). Then, the peptides were separated by means of high-performance reverse-phase liquid chromatography on a 15 cm column (Acclaim PepMap C18 100 Å, Cat. Nr. 11342013, Thermo Fischer Scientific, Waltham, MA, USA). The peptides were eluted with a linear gradient (90 min) of mobile phase B (80% acetonitrile and 0.1% formic acid) at a flow rate of 0.3 μL/min. MS analysis was performed using an Orbitrap mass-spectrometer (Q Exactive HF-X Hybrid Quadrupole-OrbitrapTM mass spectrometer, Thermo Fischer Scientific, Waltham, MA, USA) at a capillary temperature of 240 °C and an emitter voltage of 2.1 kV. Mass spectra were recorded with a resolution of 120,000 (MS) in the range of 300–1500 *m*/*z*. Tandem mass spectra of peptides were obtained through high-energy collision-induced dissociation (HCD) with a resolution of 15,000 (MS/MS) in the range from 100 *m*/*z* to the *m*/*z* value determined by the charge state of the precursor but no more than 2000 *m*/*z*.

### 4.6. Bioinformatics Data Processing

Database search for peptides and protein identification was carried out using the MaxQuant software package v2.0.1.0 [37]. The parameters of the search were the following: the human proteome database was MaxQB [38], and Trypsin/P was set as a protease with one allowed missed cleavage. The following parameters were set: variable modifications—Oxidation, Acetyl, and Deamination; fixed modifications—Carbamidomethyl; Label-free quantification (LFQ) min. ratio count—2; normalization type—classic; the first search peptide tolerance—20 ppm; min. peptide length—7; max. peptide mass—4600 Da; min. peptide length for unspecific search—8; max. peptide length for unspecific search—25; label min. ratio count—2; peptides for quantification—Unique + razor; FTMS MS/MS match tolerance—10 ppm; min. peptides for identification—2; iBAQ—“+”. The false discovery rate (FDR) using the target-decoy approach was set to 1%.

Analysis of the relative abundance of proteins in secretome fractions was performed using the Perseus software v2.0.11 [39]. ‘Only identified by site’, ‘Reverse’, and ‘Potential contaminant’ proteins, and proteins identified in less than two repeats were removed. Then, the Categorical annotation method was applied, and the resulting protein groups were compared using the Two-sample test method. Proteins with an absolute value in abundance ratio less than 2 times were filtered out. For heatmap building, the Z-score method was applied.

Single-cell RNA-Seq data for MSCs cultured in control and profibrotic conditions were obtained and processed as described in our previous paper [28]. Raw fastq-files were aligned to the reference genome human reference genome (NCBI build 38, GRCh38) using CellRanger 6.1.2 (10x Genomics, Pleasanton, CA, USA). We used the following quality control criteria: control MSCs (cells with <2000 or >7000 detected genes or <5000 or <40,000 RNA counts or over 7% unique molecular identifiers (UMIs) derived from the mitochondrial genome were excluded from further analysis), MSCs under profibrotic conditions (cells with <4000 or >8000 detected genes or <10,000 or >60,000 RNA counts or over 5% unique molecular identifiers (UMIs) derived from the mitochondrial genome were filtered out as low-quality cells). Data from samples were processed using R-studio 2023.03.1+446 (Posit, Boston, MA, USA) with R 4.2.0 and Seurat 4.1.0, regressing out mitochondrial genes (Integrated analysis of multimodal single-cell data). The integration of datasets (control MSCs and MSCs under profibrotic conditions) was performed using the Seurat function IntegrateData (default parameters). The principal component analysis of integrated datasets was performed on the variable genes, and 50 principal components were used for cell clustering (algorithm = 2 (Louvain algorithm with multilevel refinement), resolution = 0.3) and UMAP dimensional reduction. For the analysis of cluster markers, we used the function FindAllMarkers. The analysis of marker genes was performed using the g:Profiler (Gene Ontology, KEGG, Reactome, WikiPathways) (g:Profiler: a web server for functional enrichment analysis and conversions of gene lists (2019 update)). The Cell Ranger–Loupe Browser was used for visualization.

### 4.7. Western Blotting

Proteins were separated on a 12.5% SDS-PAGE gel and then transferred to a polyvinylidene difluoride membrane overnight at +4 °C at a constant voltage of 20 V using the Biorad wet system. The Toubin buffer system was used as the transfer buffer. Non-specific binding sites were blocked using 5% BSA in TBST (20 mM Tris-HCl, pH 7.6, 150 mM NaCl, and 0.1% Tween 20) for 1 h. After blocking, the membrane was incubated with primary antibodies specific for ALIX (Abcam, Cambridge, UK; dilution 1/1000), beta-tubulin (Abcam, Cambridge, UK; dilution 1/1000), fibronectin (Abcam, Cambridge, UK; dilution 1/2000), HSP70 (Biocat, Heidelberg, Germany; dilution 1/1000), CD63 (Merck Millipore, Burlington, MA, USA; dilution 1/1000), CD81 (BioLegend, San-Diego, CA, USA; dilution 1/500), collagen type I (Abcam, Cambridge, UK; dilution 1/1000), collagen type IV (Thermo Fischer Scientific, Waltham, MA, USA; dilution 1/1000), or laminin (Abcam, Cambridge, UK; dilution 1/2000) in the blocking solution overnight at +4 °C. The membrane was then washed with TBST and incubated with secondary antibodies conjugated with horseradish peroxidase (Sigma-Aldrich, St. Louis, MO, USA) in the blocking solution. Signal visualization was performed using the Clarity^TM^ Western ECL Substrate kit (BioRad, Hercules, CA, USA) on ChemiDoc (BioRad, Hercules, CA, USA).

### 4.8. qPCR

RNA isolation was performed using the ExtractRNA reagent (Evrogen, Moscow, Russia).

Reverse transcription was performed using the MMLV RT Kit (Evrogen, Moscow, Russia), and gene expression analysis through qPCR was performed using the qPCRmix-HS SYBR+LowROX reagent (Evrogen, Moscow, Russia). The primer sequences used in this work are shown in Table 3.

The differences in normalized expression levels of target genes compared to the housekeeping gene were measured through the 2^−∆∆Ct^ method using *36B4* as the normalizer gene and untreated cells as the calibrator specimen.

### 4.9. Statistical Data Processing

The experimental data are displayed as medians and interquartile ranges. The Mann–Whitney U-test was performed, and differences in results were considered statistically significant at *p* < 0.05.

## Figures and Tables

**Figure 1 ijms-25-00290-f001:**
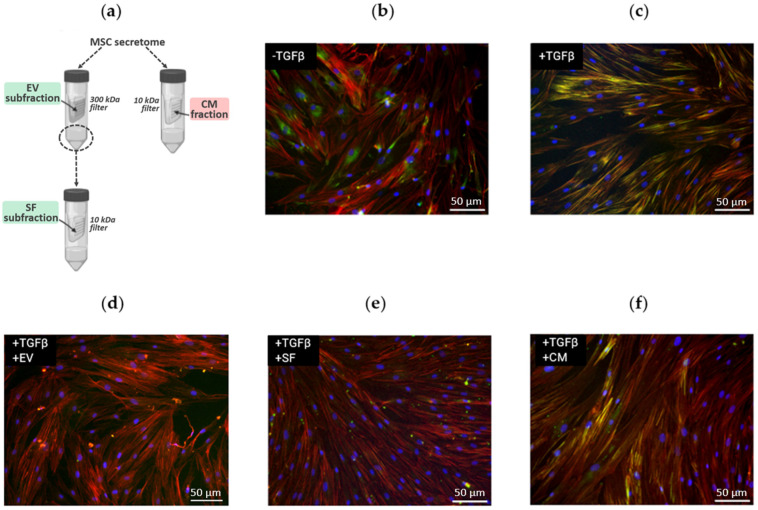
The effect of MSC secretome fractions on fibroblast-to-myofibroblast differentiation induced by TGF-β1. (**a**) Schematic representation of the isolation procedure for MSC secretome fractions. Subfractions efficiently preventing fibroblast-to-myofibroblast differentiation are marked in green, while fractions not preventing fibroblast-to-myofibroblast differentiation are marked pink. (**b**–**f**) Fluorescent microscopy of human dermal fibroblasts: (**b**) untreated (negative control); (**c**) exposed to TGF-β (positive control); (**d**) exposed to TGF-β and EV; (**e**) exposed to TGF-β and SF; (**f**) exposed to TGF-β and CM. Immunocytochemical staining for alpha-smooth muscle actin (green), F-actin staining with phalloidin (red), and nuclei staining with DAPI (blue). Stress fibers, which are characteristic of myofibroblasts, appear as yellow strands.

**Figure 2 ijms-25-00290-f002:**
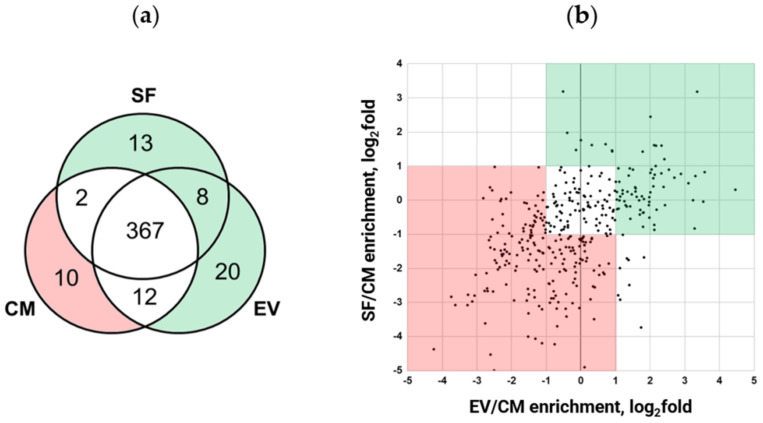
MSC secretome fractioning through ultrafiltration alters the quantitative protein content of obtained fractions. Proteins potentially mediating EV and/or SF fraction ability to prevent myofibroblast differentiation are marked green, while proteins potentially interfering with CM fraction ability to prevent myofibroblast differentiation are marked pink. (**a**) Venn diagram of secreted proteins identified in CM fraction compared to EV and SF subfractions. Pink—proteins identified in the CM and green—EV and SF fractions. (**b**) Correlation plot of the log_2_ ratios of protein abundance in EV and SF subfractions to CM fraction. Green—proteins with a relative fold of enrichment ≥ 2 and pink—proteins with a relative fold of enrichment ≤ 0.5. The combined data of three independent experiments.

**Figure 3 ijms-25-00290-f003:**
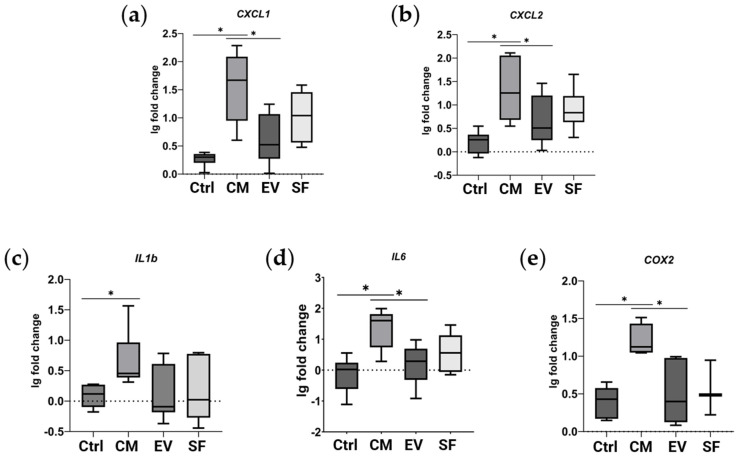
CM fraction of MSC secretome upregulates the expression of NF-κB target genes in human dermal fibroblasts. Relative gene expression compared to untreated cells measured using qPCR. Ctrl—cells treated with TGF-β1. CM, EV, and SF—cells treated with TGF-β1 and either CM fraction, EV, or SF subfraction of MSC secretome, respectively. (**a**) *CXCL1*, (**b**) *CXCL2*, (**c**) *IL1B*, (**d**) *IL6*, and (**e**) *COX2*. * indicates *p* < 0.05.

**Figure 4 ijms-25-00290-f004:**
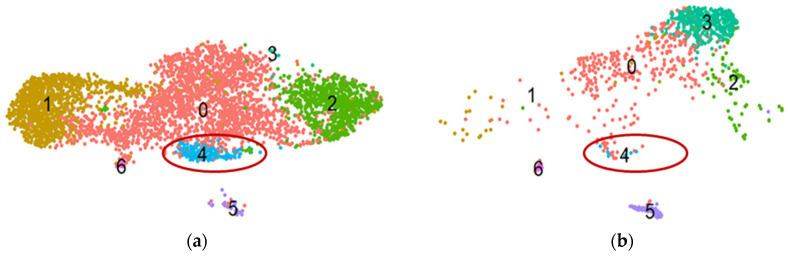
The results of scRNAseq analysis of primary human adipose-derived MSC cultures in control (**a**) and profibrotic (**b**) conditions. t-SNE plots dividing cells within cultured primary MSCs into seven clusters according to single-cell gene expression profile analysis. Notably, the MSC subpopulation (blue dots, cluster 4) with upregulated expression of genes potentially preventing myofibroblast differentiation diminishes in profibrotic conditions. (**a**) Control; (**b**) profibrotic conditions. The ellipse marks cluster 4.

**Table 1 ijms-25-00290-t001:** EV and SF subfractions are enriched in secreted proteins interacting with TGF-β, Wnt, Notch, and IGF signaling pathways compared to the CM fraction of the MSC secretome. The results of the semi-quantitative analysis of protein abundance.

Protein Name	Gene Name	CM Fraction Enrichment, Fold		Regulation of IGF Transport and Uptake by IGFBPs ^1^	Positive Regulation of NF-κB Activation ^2^
		Relative to EV Subfraction	Relative to SF Subfraction		
Aldo-keto reductase family 1 member C3	*AKR1C3*	4.4	2.8		+[11]
Disintegrin and metalloproteinase domain-containing protein 10	*ADAM10*	3.0	6.3	+	
Cadherin-2	*CDH2*	5.0	2.3	+	
Coagulation factor X	*F10*	5.4	2.9		+[12]
Protein FAM20A	*FAM20A*	8.8	6.4	+	
Stress-70 protein, mitochondrial	*HSPA9*	Detected in CM only		+	
Insulin-like growth factor-binding protein 7	*IGFBP7*	2.8	2.5	+	
Interleukin-6	*IL6*	4.9	2.5	+	+[13]
Microfibril-associated glycoprotein 4	*MFAP4*	2.9	13.2		+[14]
Macrophage migration inhibitory factor	*MIF*	18.9	26.3	+	
Matrix-remodeling-associated protein 8	*MXRA8*	2.9	2.8	+	
Plasminogen activator inhibitor 2	*SERPINB2*	3.8	6.3	+	
Superoxide dismutase [Cu-Zn]	*SOD1*	2.5	13.7	+	
Superoxide dismutase [Mn], mitochondrial	*SOD2*	2.2	Not detected in SF	+	
Spondin-2	*SPON2*	Detected in CM only			+[15]
Stromal cell-derived factor 1	*CXCL12*	5.0	3.2		+[16]
T-complex protein 1 subunit alpha	*TCP1*	Not detected in EV	3049	+	
Metalloproteinase inhibitor 1	*TIMP1*	2416	3472	+	

^1^ The data are represented in The Reactome Knowledgebase (REAC:R-HSA-381426, [10]). ^2^ The data are represented in the literature sources cited.

**Table 2 ijms-25-00290-t002:** EV and SF subfractions are enriched in secreted proteins interacting with TGF-β, Wnt, Notch, and IGF signaling pathways compared to the CM fraction of the MSC secretome. Corresponding genes are upregulated in cluster 4 relative to the total MSC population. The results of the semi-quantitative analysis of protein abundance and scRNAseq data analysis. Please see the text for further explanation.

Protein Name	Gene Name	Enrichment Relative to CM Fraction, Fold		Expression Fold Change in Cluster 4	Interacts with Signaling Pathways; Reference
		For EV Subfraction	For SF Subfraction		
Cytoskeleton-associated protein 4	*CKAP4*	Detected in EV only		1.54	Regulation of IGF transport and uptake by IGFBPs ^1^
Extracellular serine/threonine protein kinase FAM20C	*FAM20C*	4.4	0.7	-	
Growth arrest-specific protein 6	*GAS6*	Detected in SF only		(not detected)	
Interstitial collagenase	*MMP1*	0.4	2.4	(not detected)	
PRSS23	*PRSS23*	Detected in SF only		1.63	
Stanniocalcin-2	*STC2*	Detected in EV only		1.91	
Caveolin-1	*CAV1*	2.1	-	1.45	TGF-β [17]
Dickkopf-related protein 3	*DKK3*	1.5	2.1	1.26	Wnt [18]
Immunoglobulin superfamily containing leucine-rich repeat protein	*ISLR*	Not detected in CM		1.84	Wnt [19]
Latent-transforming growth factor beta-binding protein 1	*LTBP1*	5.0	3.7	-	TGF-β [20]; Regulation of IGF transport and uptake by IGFBPs ^1^
Latent-transforming growth factor beta-binding protein 2	*LTBP2*	9.4	1.2	1.69	TGF-β [21]
Matrix-remodeling-associated protein 5	*MXRA5*	2.2	0.2	1.23	TGF-β [22]
Protein NOV homolog	*NOV*	Detected in SF only		(not detected)	IGF, Notch [23]
Pappalysin-1	*PAPPA*	3.1	1.3	1.48	IGF [24]; Regulation of IGF transport and uptake by IGFBPs ^1^
Inactive tyrosine-protein kinase 7	*PTK7*	Not detected in CM		1.54	Wnt [25]
Sushi, nidogen, and EGF-like domain-containing protein 1	*SNED1*	3.5	1.3	(not detected)	Notch (predicted, [26])
Tenascin	*TNC*	11.9	2.2	1.4	TGF-β [27]; Regulation of IGF transport and uptake by IGFBPs ^1^

^1^ The data are represented in The Reactome Knowledgebase (REAC:R-HSA-381426, [10]).

**Table 3 ijms-25-00290-t003:** Primer sequences used in this study.

Gene Name	Primer Sequence (f—Forward, r—Reverse)	Source
*36B4*	f 5′-AACCGAAGTCATAGCCACAC-3′	PrimerBlast [40]
	r 5′-AACCGAAGTCATAGCCACAC-3′	
*COX2*	f 5′-ATGAGATTGTGGAAAAATTGCT-3′	[41]
	r 5′-GATCATCTCTGCCTGAGTATC-3′	
*CXCL1*	f 5′-AGTCATAGCCACACTCAAGAATGG-3′	[42]
	r 5′-GATGCAGGATTGAGGCAAGC-3′	
*CXCL2*	f 5′-AACCGAAGTCATAGCCACAC-3′	PrimerBlast [40]
	r 5′-AACCGAAGTCATAGCCACAC-3′	
*IL1B*	f 5′-AACCGAAGTCATAGCCACAC-3′	[42]
	r 5′-AACCGAAGTCATAGCCACAC-3′	
*IL6*	f 5′-AACCGAAGTCATAGCCACAC-3′	PrimerBlast [40]
	r 5′-AACCGAAGTCATAGCCACAC-3′	

## Data Availability

Data are available on request.

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
