# Peer review of "Balance between Pro- and Antifibrotic Proteins in Mesenchymal Stromal Cell Secretome Fractions Revealed by Proteome and Cell Subpopulation Analysis"

_ijms, 2023, doi:10.3390/ijms25010290_

Round 1

Reviewer 1 Report

Comments and Suggestions for Authors

In this manuscript entitled “Balance between pro- and antifibrotic proteins in mesenchymal stromal cell secretome fractions revealed by proteome and cell subpopulation analysis”, the authors compared the proteomics analysis between MSC derived secretome and extracellular vesicles or soluble factors. The authors found that protein enrichment of MSC secretome fractions and cellular subpopulations pattern shifts the balance in fibroblast-to-myofibroblasts differentiation. This manuscript provides an important contribution to the rapidly growing field of EVs; however, the quality of this manuscript is not enough for publication in its current form. Specific concerns are shown below. 

#1. In the supplementary Figure 1, the quality of the image quite low, especially CD63 and CD81. The authors should replace clear ones. 

#2. In Figure 4, the authors performed scRNAseq and showed a t-SNE plot. The authors described that under profibrotic conditions, the number of cells in cluster 4 decreased, and clusters 1 and 2 also showed a decrease. What types of cells are contained in clusters 1 and 2? In contrast, cluster 2 appears to be increasing. The authors should characterize the seven clusters.

#3. In addition to #2, the authors showed that expression fold change in cluster 4 in the table 2. It's a little confusing, so why don't you show it in a violin plot for each cluster or something? 

#4. In figure 4, the number of cells in each of the control and profibrotic condition should be shown. It should be considered whether a statistically sufficient number of cells.

Author Response

We thank esteemed Reviewer for fair criticism. We updated our manuscript according to reviewer’s comments.

#1. In the supplementary Figure 1, the quality of the image quite low, especially CD63 and CD81. The authors should replace clear ones. 

We replaced staining for cd63 and cd81 and also added  staining for vesicular protein ALIX.

 #2. In Figure 4, the authors performed scRNAseq and showed a t-SNE plot. The authors described that under profibrotic conditions, the number of cells in cluster 4 decreased, and clusters 1 and 2 also showed a decrease. What types of cells are contained in clusters 1 and 2? In contrast, cluster 2 appears to be increasing. The authors should characterize the seven clusters.

 We added characterization of each of seven clusters to the text (lines 209-222).

Cluster 0 includes cells with upregulated genes associated with the actin cytoskeleton regulation and formation of tight junctions (e.g., TPM2, ACTA2, CALD1, TAGLN). Cluster 1 corresponds to MSCs in a basal undifferentiated state; сells of this cluster are characterized by increased expression of genes associated with oxidative phosphorylation and synthetic processes (SNHG29, RPS12, ATP5F1E, HSPB7, HSPB6, ATP5MD, RPL12, RPS27L). Cluster 2 includes cells in the G2M and S phases of the cell cycle; сells of this cluster express genes associated with the cell cycle (TOP2A, TYMS, MKI67, CENPF, TUBB4B, TUBA1B, PTTG1). Cluster 3 represents cells with increased expression of genes associated with extracellular matrix degradation (MMP2, CTSK, LUM, CLU). Cluster 4 includes cells expressing extracellular matrix genes (POSTN, COL5A1, FBN1). Cluster 5 cells are characterized by increased expression of genes associated with contractility (MYH11, ACTA2, ACTG2, MYLK, CALD1) and, therefore, potentially displaying smooth muscle cell-like phenotype. Cluster 6 cells are characterized by the expression of genes associated with the regulation of the immune response (CXCL8, LYZ, CCL2, CCL3, CD74, FCER1G).

#3. In addition to #2, the authors showed that expression fold change in cluster 4 in the table 2. It's a little confusing, so why don't you show it in a violin plot for each cluster or something? 

We added violin plots to supplementary figures (Figure S4) and added notion in the text (lines 234-235).

#4. In figure 4, the number of cells in each of the control and profibrotic condition should be shown. It should be considered whether a statistically sufficient number of cells.

We have 4438 cells in control sample and 929 cells in “Profibrotic conditions” sample. To analyze the proportion of cells in each cluster, we first took a number of cells of a single sample in the cluster and then divided that number by the total number cells of that sample. We presented data on number and percent of cells in each cluster for both Control and "Profibrotic conditions" sample in Supplementary table S4. 

Cluster

Percent of cells in cluster of Control sample

Percent of cells in cluster of "Profibrotic conditions" sample

Number of cells in cluster

Number of cells of Control sample in cluster

Number of cells of "Profibrotic conditions" sample in cluster

0

48.2

36.81

2481

2139

342

1

28.23

3.66

1287

1253

34

2

18.77

7.21

900

833

67

3

0.14

41.01

387

6

381

4

3.85

0.86

179

171

8

5

0.59

8.29

103

26

77

6

0.23

2.15

30

10

20

Reviewer 2 Report

Comments and Suggestions for Authors

This is a well written manuscript, authors presented compelling data suggesting that differential fractionation of MSCs may impact their therapeutic efficacy in the fibrosis content. They identified different proteome signature in various fractions of MSCs that modified their anti-fibrotic properties in vitro.

The focus of the study was to identify secereted factors/EV cargos that are involved in dual action (profibrotic vs anti-fibrotic) of mesechymal stromal cells (MSC) in fibrotic conditions.

Although, there are studies performing protemics analysis of MSC conditioned medium or MSC-EV's content, to my knowledge, this is the first study to compare the proteome signature of different fraction of MSC and to study differential impact of each fraction in progression of fibrosis. So, I believe that the study has brought new findings to the field.

The method section has been explained clearly and sufficiently, although adding the antibodies dilution used for immunohistochemistry and Werstern blot experiment would be useful for other investigators wishing to use the same antibodies.    The experiments design were satisfactory.

The conclusions are consistent with the evidence and arguments presented.

Data are clearly presented, methods are well explained and discussion is satisfactory. I would only request a minor edit to add a reference in the discussion section, 286-287 where they stated that "Also, like many other cells of mesenchymal origin, MSCs itself are able to differentiate into myofibroblasts after the stimulation by TGF-β."

Author Response

We thank esteemed Reviewer for fair criticism. We updated our manuscript according to reviewer’s comments.

The method section has been explained clearly and sufficiently, although adding the antibodies dilution used for immunohistochemistry and Western blot experiment would be useful for other investigators wishing to use the same antibodies.

We provided additional information on antibody dilution to the Methods section (lines 370 and 453-457).

Data are clearly presented, methods are well explained and discussion is satisfactory. I would only request a minor edit to add a reference in the discussion section, 286-287 where they stated that "Also, like many other cells of mesenchymal origin, MSCs itself are able to differentiate into myofibroblasts after the stimulation by TGF-β."

The main source of myofibroblasts are tissue resident fibroblasts, however some of myofibroblasts originate from other cells of mesenchymal origin such as pericytes, smooth muscle cells, MSCs and others. We clarified that point in the text (303-304).

Round 2

Reviewer 1 Report

Comments and Suggestions for Authors

 I checked revised version. All of my concern have been revised.